# Meiotic DNA breaks activate a streamlined phospho-signaling response that largely avoids protein-level changes

Funda M Kar, Christine Vogel , Andreas Hochwagen

**Meiotic cells introduce a numerous programmed DNA breaks into their genome to stimulate meiotic recombination and ensure controlled chromosome inheritance and fertility. A checkpoint network involving key kinases and phosphatases coordinates the repair of these DNA breaks, but the precise phosphorylation targets remain poorly understood. It is also unknown whether meiotic DNA breaks change gene expression akin to the canonical DNA-damage response. To address these questions, we analyzed the meiotic DNA break response in *Saccharomyces cerevisiae* using multiple systems-level approaches. We identified 332 DNA break–dependent phosphorylation sites, vastly expanding the number of known events during meiotic prophase. Less than half of these events occurred in recognition motifs for the known meiotic checkpoint kinases Mec1 (ATR), Tel1 (ATM), and Mek1 (CHK2), suggesting that additional kinases contribute to the meiotic DNA-break response. We detected a clear transcriptional program but detected only very few changes in protein levels. We attribute this dichotomy to a decrease in transcript levels after meiotic entry that dampens the effects of break-induced transcription sufficiently to cause only minimal changes in the meiotic proteome.**

## Introduction

Homologous recombination during meiosis is initiated by programmed DNA breaks created by the conserved topoisomerase-like protein Spo11 (Bergerat et al, 1997; Keeney et al, 1997). By stimulating crossover formation, these DNA breaks promote genetic diversity in the offspring and, in many organisms, are essential for faithful segregation of chromosomes (Lam & Keeney, 2015). Accordingly, failure to produce DNA breaks leads to infertility in yeast and mammals (Klapholz et al, 1985; Baudat et al, 2000; Romanienko & Camerini-Otero, 2000). However, the large number of DNA breaks also represents an inherent hazard for genome instability. To

protect the genome, a complex signaling network coordinates meiotic processes in response to DNA break formation and prevents inappropriate meiotic progression when break repair is delayed or defective (MacQueen & Hochwagen, 2011; Subramanian & Hochwagen, 2014).

We are only beginning to understand the signaling pathways that connect meiotic DNA break formation to DNA repair and other meiotic processes. Available data indicate a prominent role for the canonical DNA-damage sensor kinases ATR and ATM, although the extent to which the two kinases are linked to the control of meiotic processes may vary between organisms (Kar & Hochwagen, 2021). In the budding yeast, *Saccharomyces cerevisiae*, the homologues of ATR and ATM, Mec1, and Tel1, respectively, and the downstream CHK2-like effector kinase Mek1 regulate a large number of meiotic processes, including DNA break formation and repair, chromosome pairing, and meiotic cell-cycle progression (Kar & Hochwagen, 2021). Targeted studies in yeast have identified relevant DNA break–dependent phosphorylation events for several of these processes, including control of break levels (Carballo et al, 2013), suppression of sister-chromatid recombination (Carballo et al, 2008; Niu et al, 2009; Callender et al, 2016), crossover formation (Chen et al, 2015; He et al, 2020; Woo et al, 2020), centromere uncoupling (Falk et al, 2010), and control of meiotic cell-cycle progression (Penedos et al, 2015; Chen et al, 2018). Targeted mutagenesis based on known kinase motifs mapped additional, functionally important break-dependent phosphorylation sites (Bartrand et al, 2006; Cartagena-Lirola et al, 2006; Serrentino et al, 2013), and a number of DNA break–dependent, site-specific phosphorylation events of unknown functional significance have been identified (Shroff et al, 2004; Suhandynata et al, 2016; Kniewel et al, 2017). However, available data (MacQueen & Roeder, 2009; Garcia et al, 2015; Mohibullah & Keeney, 2017) indicates that our understanding of the signaling response to meiotic DNA break formation is far from complete.

One little-explored aspect of the meiotic DNA break response is the relationship between the signaling pathways discussed above and changes in gene expression. The canonical DNA-damage response in vegetative yeast cells involves a well-defined transcriptional response that is signaled through Mec1/Tel1-dependent

Department of Biology, New York University, New York City, NY, USA

Correspondence: andi@nyu.edu; cvogel@nyu.edu

activation of the effector kinases Rad53 and Dun1 (Jaehnig et al, 2013). These kinases activate a core set of DNA-damage response genes, including genes coding for DNA-repair factors and regulators of nucleotide abundance. Rad53 activity is attenuated, but not absent, during meiotic DNA break formation (Cartagena-Lirola et al, 2008; Falk et al, 2010), posing the question whether transcriptome and proteome remodeling has a role in the meiotic DNA break response.

To address these questions, we analyzed the cellular response to meiotic DNA breaks with respect to system-wide phospho-proteomic, proteomic, and transcriptomic changes. We identified hundreds of novel DNA break–dependent phosphorylation sites, highlighting the breadth of the signaling pathways in response to meiotic DNA breaks. Surprisingly, we observed a substantial transcriptional response but only minimal changes in the proteome.

# Results

### Experimental setup to measure multiple aspects of the meiotic DNA break response

To capture the meiotic DNA break response from multiple angles, we performed transcriptomic, proteomic, and phospho-proteomic analyses in *S. cerevisiae* (Fig 1A). We compared DNA break–proficient cells carrying a functional *SPO11* gene with catalytic *spo11-YF* mutants, which cannot form DNA breaks (Bergerat et al, 1997). To ensure the most robust and stringent analysis possible, we implemented several additional experimental features. First, to avoid differences in meiotic state between DNA break competent *SPO11* cells, which delay in prophase compared with *spo11* mutants (Keeney, 2001), we removed the mid-meiosis transcription factor Ndt80 genetically from both strains. *NDT80* deletion synchronizes

both cultures before the exit from meiotic prophase (Xu et al, 1995), and thus eliminates cell-cycle differences, which create a well-known false-positive signal when studying the cellular response to DNA damage (Gasch et al, 2001; Suhandynata et al, 2016). Second, to increase recovery of DNA break–dependent phosphorylation events, we used a *pph3Δ* mutation to inactivate protein phosphatase 4 (PP4), one of the major phosphatases responsible for erasing Mec1/Tel1-dependent phosphorylation marks (Keogh et al, 2006; Falk et al, 2010; Hustedt et al, 2015). Finally, to ensure reproducibility, proteomics and phospho-proteomics samples were collected from three independent biological replicates and analyzed by two complementary mass spectrometry techniques to maximize recovery of phosphosites. We used flow cytometric analysis of DNA replication as a proxy for meiotic synchronization to show that meiotic cultures completed S phase with similar kinetics within each set of biological replicates (Fig S1).

We quantified tryptic peptides before and after phospho-peptide enrichment to map DNA break–dependent changes in protein abundance and protein phosphorylation, respectively (Fig 1A). We used both data-dependent acquisition (DDA) and data-independent acquisition (DIA) to identify and quantify phosphorylation sites. DDA is the traditional method but is semi-stochastic as it only identifies the most abundant peptides. This filter biases the data towards high-abundance peptides and introduces variation in identified peptides between runs (Domon & Aebersold, 2010; Michalski et al, 2011). DIA overcomes this problem by co-fragmenting all of the peptides in pre-defined mass/charge windows. The resulting fragmentation spectra are highly complex and require more advanced algorithms and spectral libraries to resolve peptide sequences (Schubert et al, 2015), but DIA can achieve greater reproducibility and quantitative sensitivity than DDA (Selevsek et al, 2015; Bruderer et al, 2017), in particular for phosphoproteomics (Bekker-Jensen et al, 2020; Kitata et al, 2021).

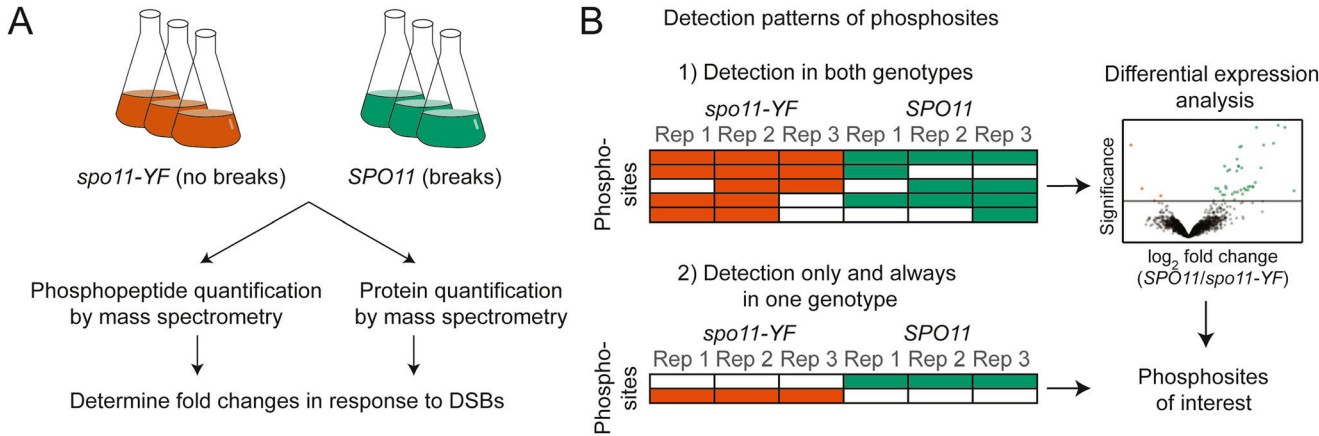

**Figure 1. Workflow and analysis setup.**
**(A)** We extracted proteins from synchronous meiotic cultures and quantified the phosphopeptides and proteins obtained from *SPO11* and *spo11-YF* cells. Proteomics and phosphoproteomics experiments both had three biological replicates. **(B)** Quantitative and qualitative strategies were used to classify phosphorylation sites of interest. Filled rows represent non-zero intensity values for the phosphosites. When there were enough non-zero intensity values, differential expression analysis was used to classify DNA break–dependent sites or phosphorylation sites that were lost in response to DNA breaks (quantitative approach). In addition, we used a qualitative approach by analyzing detection patterns for phosphosites. When a phosphosite was present in all replicates of *SPO11* and absent in all *spo11-YF* samples, it was designated as a DNA break–dependent site. If a site was present in all replicates of *spo11-YF* and absent in all *SPO11* samples, it was categorized as lost in response to DNA breaks. DSB, double strand breaks; Rep, replicate.

We designed a workflow that includes both methods to take advantage of their unique strengths.

### Meiotic DNA break response involves hundreds of phosphorylation events

We found good reproducibility across all three biological replicates for both the DDA and DIA proteomic analysis, and both methods efficiently recovered DNA break–dependent phosphorylation sites. We observed some loss of phosphorylation events in replicate 1, either during sample preparation or because of biological variability, as indicated by the narrow distribution phosphopeptide intensity differences between *SPO11* and *spo11-YF* cells in both the DDA and DIA analysis (Fig S2A and C). Inspection of DNA content profiles showed that cultures from replicate 1 replicated their DNA slightly faster (Fig S1), which may have resulted in longer prophase residence time, and led to loss of phosphorylation on some sites. This loss resulted in lower correlation of this replicate with the other replicates when examining all identified sites (Fig S2B and D). However, the correlation with the other replicates was strong when considering only phosphosites that were likely true-positive identifications, that is, with an increased intensity in *SPO11* samples, suggesting this replicate produced meaningful data (Fig S3). In

addition to this, replicate 1 successfully reported previously characterized DNA break–dependent phosphorylation events such as Zip1 S75 and Hed1 T40 (Falk et al, 2010; Callender et al, 2016), further confirming its validity. Therefore, we chose to retain the data from all three replicates. To ensure high-quality data, we removed phosphosite identifications with more than three values missing from the six measurements of DNA break dependence.

We used two different approaches to establish the phosphosites affected by DNA break formation (Fig 1B). First, we extracted phosphosites detected in both *SPO11* and *spo11-YF* samples in at least three of the six samples and determined relative enrichment in *SPO11* versus *spo11-YF*. This analysis classified 40 and 157 phosphorylation sites as significantly enriched in the presence of meiotic DNA breaks (adjusted *P*-value cutoff 0.1) in the DDA and DIA data, respectively (Fig 2A and B). Although these phosphorylation events were induced by DNA breaks, their presence in *spo11-YF* samples suggested that they can occur independently of meiotic DNA breaks as well.

Second, we conducted a presence/absence analysis to also include phosphosites that were never detected in *spo11-YF* samples, that is, phosphosites that are likely fully dependent on meiotic DNA break formation (Fig 1B). This analysis revealed another 118 and 84 sites as phosphorylated only and always in presence of meiotic DNA breaks in the DDA and DIA data, respectively. In total,

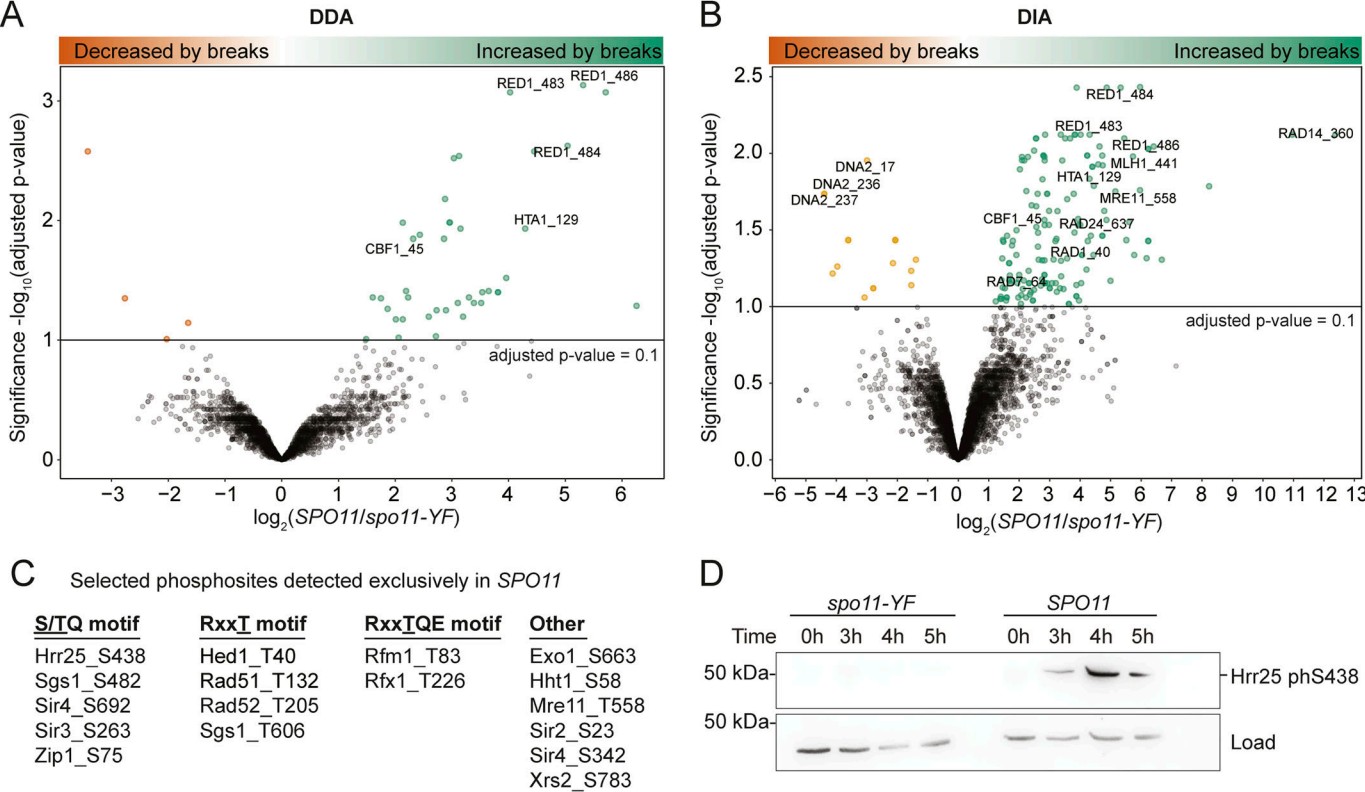

**Figure 2. Changes in protein phosphorylation in response to meiotic DNA breaks.**
**(A)** A volcano plot showing results of differential expression analysis for data-dependent acquisition data, each dot representing a phosphosite. −log₁₀ (Benjamini–Hochberg adjusted *P*-value) is shown on the y-axis and log₂ fold change is shown on the x-axis. Labels indicate the phosphorylation sites on peptides. **(B)** A volcano plot showing results of differential expression analysis for data-independent acquisition data, each dot representing a phosphopeptide. **(C)** A table listing selected DNA break–dependent phosphorylation sites discovered by the presence/absence analysis (see Table S1 for the complete list). **(D)** A Western blot showing DNA break dependence of Hrr25 S438 phosphorylation. β-actin was used as a loading control. DDA, data dependent acquisition; DIA, data independent acquisition.

DDA and DIA analysis resulted in 158 and 241 DNA break–dependent phosphosites, respectively, with 67 sites captured by both methods.

We examined the method-specific site identifications and found that most were explained by missing data or scores below the significance cutoff (Fig S4). Based on the results of these analyses, we opted to merge all DNA break–dependent phosphosites from the DDA and DIA datasets, obtaining a total of 332 DNA break–dependent phosphosites (Table S1). In addition, we detected 4,953 phosphorylation events that were not DNA break dependent.

Among the DNA break–dependent sites were several previously confirmed Mec1/Tel1 targets, including H2A S129, Cbf1 S45, Zip1 S75, Rad54 T132, and Hed1 T40 (Downs et al, 2000; Smolka et al, 2007; Niu et al, 2009; Falk et al, 2010; Callender et al, 2016) (Fig 2A–C), confirming the high quality of our dataset. In addition, we uncovered many DNA break–dependent phosphorylation events that have only been characterized in non-meiotic cells or that are entirely novel. For example, we identified an additional DNA break–dependent phosphorylation event on histone H3 S57 (Fig 2C), which is predicted to weaken nucleosomal DNA association (Bowman & Poirier, 2015), and may thus play a role in removing nucleosomes during meiotic DNA repair. We also identified DNA break–dependent phosphorylation on several chromatin factors, including all three subunits of the Sir complex (Sir2, Sir3, Sir4), which have roles in meiotic checkpoint regulation and the timing of meiotic prophase (San-Segundo & Roeder, 1999; Subramanian et al, 2019) (Fig 2C). Among proteins involved in meiotic DNA repair, we identified additional DNA break–dependent phosphorylation sites on the MRX complex (Mre11 and Xrs2), regulators of recombination (Rad52 and Sgs1) and the meiotic resolvase complex (Mlh1 and Exo1; Fig 2B and C). Intriguingly, we also observed DNA break–dependent phosphorylation of multiple components of the nucleotide-excision repair machinery (Rad1, Rad7, Rad14, Rad16, Rad23, and Rad26; Fig 2B). As there is no known role for nucleotide-excision repair during meiotic prophase, these phosphorylation events may be inhibitory.

We also observed a total of 81 phosphosites that disappeared during DNA break formation, as they were either enriched in *spo11-YF* compared with *SPO11* samples or were specifically detected in *spo11-YF* samples (Table S2). Among these sites were Cdk1-dependent phosphorylation events on Dna2 S17 and S237 (Fig 2B), which are known to regulate Dna2 localization to DNA breaks and may promote Mec1/Tel1-dependent phosphorylation of yet unidentified sites on Dna2 (Chen et al, 2011). Thus, the observed DNA break–dependent reduction in abundance of some of these phosphorylations is likely biologically meaningful.

Several known DNA break–dependent sites were absent from our data. Specifically, neither DDA nor DIA data captured Hop1 T318 or H3 T11 (Carballo et al, 2008; Kniewel et al, 2017). Inspection of the sequences surrounding these sites revealed that the H3 T11 phosphorylation site would reside on a tryptic peptide consisting of only five amino acids, which is too small for reliable proteomic identification. Conversely, a tryptic peptide with Hop1 T318 phosphorylation would be 49 amino acids long; and larger peptides are typically detected at a low frequency (Swaney et al, 2010).

Finally, we validated one newly identified site, namely the phosphorylation of the yeast casein kinase $1\delta/\varepsilon$ homologue Hrr25 on S438 by raising and purifying a phospho-specific antibody (Fig S5). Immunoblotting showed that Hrr25 S438 occurred specifically in response to meiotic DNA break formation and was undetectable in meiotic *spo11-YF* cultures (Fig 2D). Therefore, our data present a high-quality, rich resource for phospho-signaling during the meiotic DNA break response.

## Phosphorylation sites are enriched for known kinase motifs

To better define substrate selection in response to meiotic DNA break formation, we examined features of DNA break–dependent phosphosites in our data. Motif enrichment analysis showed strong enrichment (motif-x, *P*-value cutoff 0.05) of predicted consensus sites for Mec1/Tel1 (S/TQ) and Mek1 (RxxT), the main regulators of the meiotic DNA break response. Approximately 19% of all DNA break–dependent sites occurred in an S/TQ motif, compared with only ~5% S/TQ occurrence among all phosphorylation sites detected in our study (Fig 3A). Among DNA break–dependent S/TQ sites, L was the most common amino acid at the −1 position and E was the most common amino acid at the +2 position (Fig 3B). As LS/TQE is known as a high-affinity site for human ATM in in vitro studies (O'Neill et al, 2000), these sites are likely also targeted by the yeast orthologs Mec1/Tel1.

Approximately 28% of all DNA break–dependent sites localized to RxxS/T motifs, compared with only ~17% of all detected sites (Fig 3A). Among these sites, we observed S as the most common amino acid at the −2 position (Fig 3C). The RxxS/T motif is preferred by several kinases in yeast (Mok et al, 2010), but all known Mek1-dependent phosphorylation sites contain a threonine instead of a serine (Niu et al, 2009; Callender et al, 2016; Kniewel et al, 2017). Indeed, among the detected phosphorylation sites with an RxxS/T motif, DNA break–dependent sites were significantly enriched for threonine as the phosphorylated amino acid (hypergeometric test, *P*-value <0.001; Fig S6A), consistent with the reported sequence preference of Mek1 (Suhandynata et al, 2016).

In a couple of instances, we also observed phosphorylation of RxxTQE sequences (Fig 2C), which combine RxxT and TQE motifs and may thus represent a hybrid target motif for Mek1 and Mec1/Tel1. As Mek1 is activated by Mec1/Tel1, phosphorylation of RxxTQE hybrid motifs by both types of kinases could create a coherent feedforward signal, a common feature in gene regulatory networks that creates signal stability (Mangan & Alon, 2003). Notably, the two proteins with phosphorylated RxxTQE motifs in our data, Rfm1, and Rfx1, are transcription factors of the DNA damage response, with Rfm1 preventing premature exit from meiotic prophase (Xie et al, 1999; McCord et al, 2003).

Although our data show a clear enrichment of predicted Mec1/Tel1 and Mek1 motifs, more than half (~53%) of all DNA break–dependent phosphosites had neither motif, suggesting other kinases might be catalyzing these phosphorylation events. Motif search failed to identify additional significantly enriched motifs among these sites, possibly because of a lack of sequence preference or because of contributions from multiple kinases. Candidate kinases include Rad53 and Dbf4-dependent kinase (DDK), both of which have been shown to mediate phosphorylation events downstream of Mec1/Tel1/Mek1 and/or meiotic DNA break formation (Bashkirov et al, 2003; Chen et al, 2015; Lao et al, 2018; He et al, 2020), with only DDK having a known consensus motif.

In general, DNA break–dependent phosphorylation events localized to intrinsically unstructured parts of target proteins (Fig S6B), presumably reflecting increased accessibility and the higher preponderance of serines and threonines in these regions. This

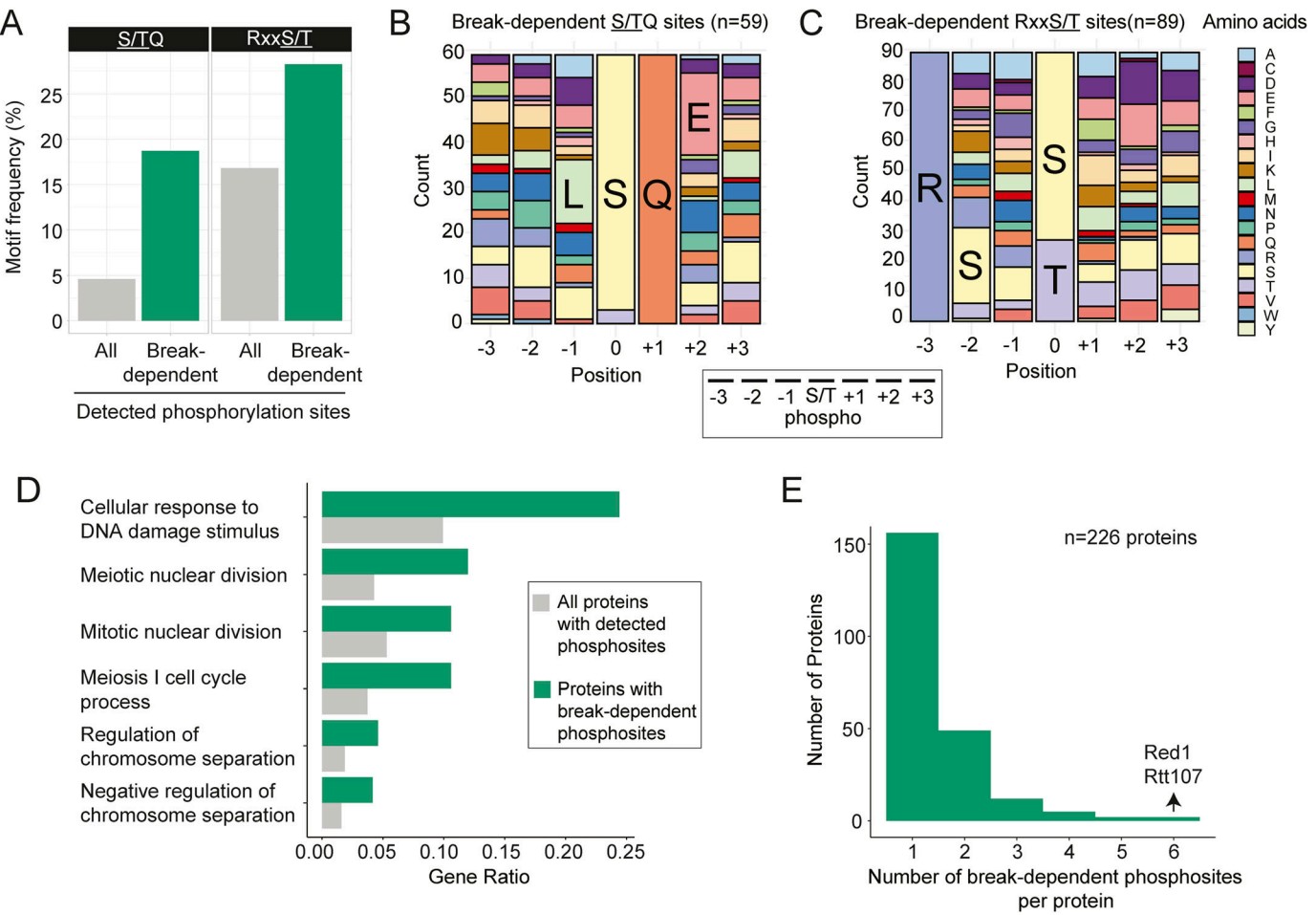

**Figure 3. Characteristics of DNA break–dependent phosphorylation events.**
**(A)** DNA break–dependent sites were enriched for Mec1/Tel1 (S/TQ) and Mek1(RxxT) consensus motifs, compared with all phosphorylation sites detected in our study. **(B)** Stacked bar graph representing the distribution of amino acids surrounding the DNA break–dependent S/TQ sites. The most frequent amino acids at the −1 and +2 position were leucine (L) and glutamic acid (E), respectively. **(C)** Stacked bar graph representing the distribution of amino acids surrounding the DNA break dependent RxxS/T sites. Serine (S) at the −2 position was the most frequent amino acid, whereas there was no striking feature for the other positions. **(D)** Bar graphs showing the results of functional enrichment analysis of proteins with DNA break–dependent phosphosites. **(E)** Bar graph showing the distribution of detected DNA break–dependent phosphorylation sites per protein.

trend is not unique to DNA break–dependent phosphorylation events (Fig S6B) and has also been noted in previous large-scale phospho-proteomics analyses (Holt et al, 2009). Interestingly, more than one third (116 of 332) of DNA break–dependent phosphorylation events occurred within five amino acid of another DNA break–dependent site, which may either reflect multiple phosphorylation events mediated by the same kinase, or priming events, whereby phosphorylation by one kinase stimulates nearby phosphorylation events by another kinase.

The 332 phosphorylation sites mapped to a total of 226 different proteins which were strongly enriched for functions related to meiosis and DNA repair (q-value < 0.2, Fig 3D), implying that many of these phosphorylation events likely are functional. For most proteins, we only identified a single DNA break–dependent site, whereas two proteins had six DNA break–dependent phosphorylation sites (Fig 3E). One of these proteins is the meiotic chromosome organizer Red1, which has been suggested to be phosphorylated in both DNA

break–dependent and –independent manner (Bailis & Roeder, 1998; de los Santos & Hollingsworth, 1999; Lai et al, 2011; Wan et al, 2004). Consistent with this notion, our data shows 10 additional phosphorylation sites for Red1 that are not DNA break dependent. One of the DNA break–dependent sites on Red1, T484, fits the consensus motif for phosphorylation by Mek1, but phosphorylation of this site was still detectable at low levels in *spo11-YF* strains indicating that it is not solely dependent on meiotic DNA break formation.

## Meiotic DNA breaks do not trigger major proteome changes

We tested whether the meiotic DNA break response also affects protein levels. To this end, we quantified proteins in the same replicate cultures described above. This analysis yielded robust quantitative data for proteins from just under half of the protein-coding genes in yeast (2,627 proteins), and thus likely captured a major proportion of the meiotically expressed proteome. Measurements were highly

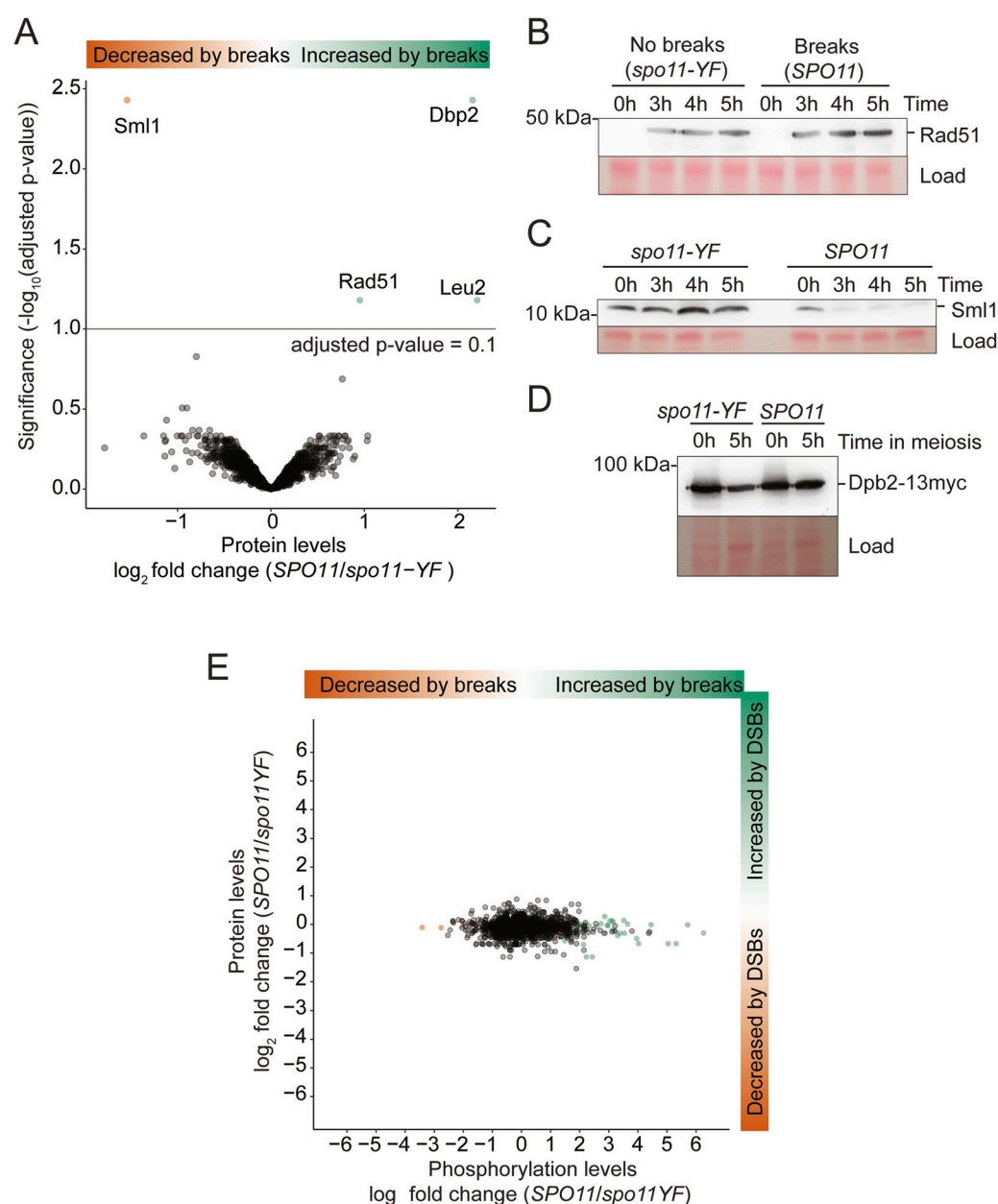

**Figure 4. Few proteins change significantly in abundance in response to meiotic DNA breaks.**
**(A)** A volcano plot showing differentially expressed proteins, each dot represents a protein. The y-axis shows −log$_{10}$ (Benjamini & Hochberg adjusted *P*-value) and the x-axis shows log$_2$ fold change. **(B)** Western blot analysis of Rad51 in *spo11-YF* and *SPO11* cells during meiosis is shown. Ponceau S was used as the loading control. **(C)** Comparison of Sml1 levels in *spo11-YF* and *SPO11* cells during meiotic prophase. Sml1 was diminished after 3 h into meiosis concomitantly with DNA break formation. Ponceau S staining of the membrane was the loading control. **(D)** Assessment of Dbp2 levels by Western blotting before meiosis (0 h) and during meiotic prophase (5 h) is shown. Dbp2 was tagged with a 13xmyc tag and anti-myc antibody was used for detection of Dbp2. Ponceau S was the the loading control. **(E)** Plot showing correlation between protein-level log$_2$ fold changes (y-axis) and phosphosite-level log$_2$ fold changes (x-axis). Each dot represents a phosphosite. Whereas log$_2$ fold changes for phosphosites distributed widely, protein log$_2$ fold changes for proteins centered narrowly around 0.

correlated between replicates and samples (r = 0.99 between replicates; Fig S7A–C). Surprisingly, log$_2$ fold changes in response to DNA break formation were narrowly centered on zero similar to those between replicate measurements (Fig S8A), suggesting that meiotic DNA breaks do not change the proteome to a detectable extent.

We observed only four proteins with significantly changed abundance between *SPO11* and *spo11-YF* cultures (adjusted *P*-value

< 0.1): the levels of Rad51, Leu2, and Dbp2 were elevated in *SPO11* cells; Sml1 levels were decreased (Fig 4A). We validated the relative changes in Rad51, Sml1, and Dbp2 levels by immunoblotting (Fig 4B–D). Rad51 and Sml1 are known targets of the DNA damage response. Rad51 is a recombinase required for DNA break repair that is induced upon DNA damage and also protected from degradation (Basile et al, 1992; Woo et al, 2020), whereas the ribonuclease

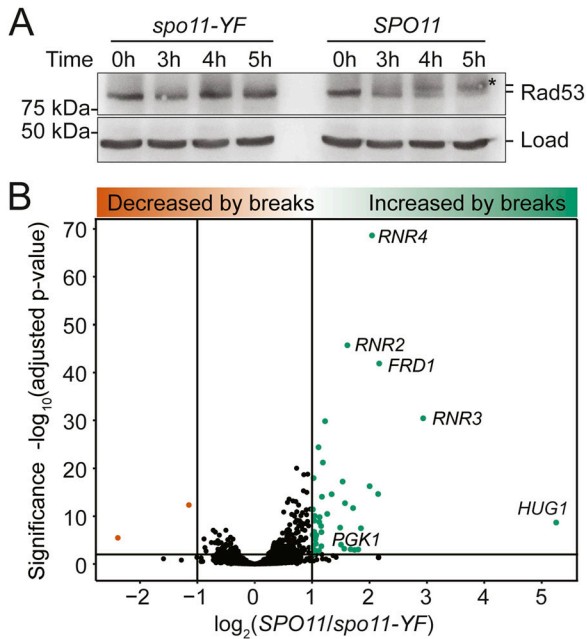

**Figure 5. Activation of a transcriptional program in response to meiotic DNA breaks.**
**(A)** Western blot analysis of Rad53 in *spo11-YF* cells and *SPO11* cells. We used the mobility shift of Rad53 as a proxy for its phosphorylation and activation. A cross-reacting band produced by the Rad53 antibody was the loading control. Note: this analysis was performed on our experimental strains, which lack Pph3 phosphatase, resulting in more easily detectable Rad53 autophosphorylation than seen in strains with active Pph3 (Cartagena-Lirola et al, 2008; Falk et al, 2010). **(B)** A volcano plot showing the results of mRNA-Seq analyses of *spo11-YF* cells and *SPO11* cells. The y-axis shows -log$_{10}$ (Benjamini–Hochberg adjusted *P*-value) and the x-axis shows log$_2$ fold change between *SPO11* and *spo11-YF*. Each dot represents a gene.

reductase RNR inhibitor Sml1 is degraded (Zhao, 2001; Andreson et al, 2010). The resulting changes in abundance confirmed our mass spectrometry data (Fig 4A–C), indicating that these aspects of the DNA damage response remain active in response to meiotic DNA breaks. Differences in Leu2 levels were expected given that *SPO11* cells contained four copies of the *LEU2* gene whereas the *spo11-YF* cells only contained two (Table S3). The significantly higher Leu2 levels in the *SPO11* strain thus served as an internal control verifying the quality and sensitivity of the measurements. Dbp2 is an essential RNA-binding protein without known function during meiosis or the DNA damage response. When testing its meiosis-specific depletion, we observed only a minor loss in gamete viability (Fig S9). Thus, although it is possible that some undetected proteins did change in expression, our data indicate that the meiotic DNA break response does not remodel the proteome at a large scale. This result also means that the observed phosphorylation changes were not driven by changes in protein abundance (Figs 4E and S8B).

### Activation of the signaling pathway for damage-dependent transcription

The robustness of the proteome in response to meiotic DNA break formation was surprising, as it contrasted with the transcript and protein abundance changes that are characteristic of the canonical DNA damage response (Elledge & Davis, 1990; Huang et al, 1998; Gasch et al, 2001; Tsaponina et al, 2011; Jaehnig et al, 2013). We therefore investigated whether the effector kinases Rad53 and Dun1, which trigger the DNA damage–dependent transcription changes, were active during the response to meiotic DNA breaks. Western blotting of Rad53 in our experimental strains showed phosphorylated, slower migrating forms of Rad53 only in *SPO11* cells but not in *spo11-YF* cells (Fig 5A), indicating that Rad53 is activated by meiotic DNA breaks. In addition, we detected DNA break–specific phosphorylation of Dun1 at position S10 (Chen et al, 2007) (Table S1) and reduction of Sml1 protein levels (Fig 4A and C), both of which are hallmarks of Dun1 activation, as Sml1 is a Dun1 target (Zhao & Rothstein, 2002). Thus, key regulators of the transcriptional response to canonical DNA damage are activated also in response to meiotic DNA break formation.

### A transcriptional response to meiotic DNA breaks

We tested if activated Rad53 and Dun1 induce transcription of DNA damage-response genes during meiosis. To this end, we conducted mRNA-seq experiments using the same strains as described above, comparing mRNA levels of *SPO11* and *spo11-YF* cultures. We found that 7% of genes (373 of 5,386) were differentially expressed (adjusted *P*-value < 0.01), and 42 genes (<1%) were up-regulated more than twofold in response to DNA break formation. This group contained many genes of the canonical gene expression response to DNA damage, including *RNR2*, *RNR3*, and *RNR4* (Elledge & Davis, 1987, 1990; Huang & Elledge, 1997) (Fig 5B). This result was unexpected because the DNA damage–dependent transcriptional induction in mitotic cells directly results in a clear increase in protein levels (Yao et al, 2003; Huang et al, 2016).

We conducted several tests to confirm that, indeed, the meiotic proteome is largely fixed despite DNA break–dependent transcriptome remodeling. We verified that the discordance could not be explained by lower coverage of the proteome data. Of the 287 transcriptionally up-regulated genes, 165 (57%) were quantified in the proteome data, including 23 (54%) of the 42 genes with >2-fold change at the RNA level, suggesting that the proteomics experiment captured a representative fraction of the proteome. Importantly, protein levels of the latter group only increased by less than 10%, which, although statistically significant compared to the rest of the proteome (Fig S10A), was far below the changes at the mRNA level. Immunoblotting further confirmed the mass spectrometry based results: the increased levels of *RNR4*, *FRD1*, and *PGK1* transcripts detected by mRNA-seq (Fig 5B) only caused minor DNA break–dependent changes in protein levels (Figs 6A and B and S10B). Therefore, we conclude that, unlike in the mitotic DNA damage response, the transcriptional changes in response to meiotic DNA breaks do not lead to major changes in protein abundance.

### Meiotic entry associates with strong reduction in mRNA abundance

To further investigate these results, we used Northern blotting to analyze the transcript levels of several genes in a meiotic time course. We observed two competing effects on mRNA levels. All analyzed transcripts experienced a noticeable decrease in abundance as

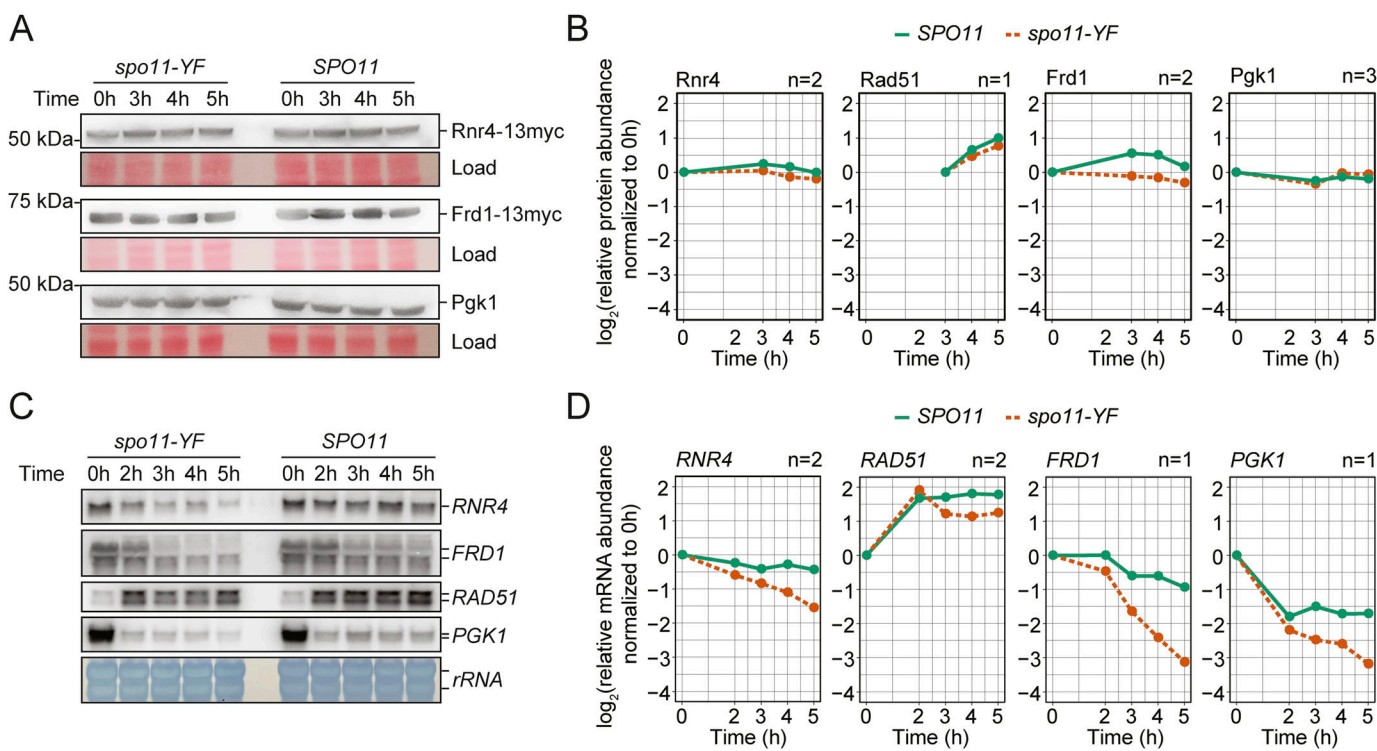

**Figure 6. Activation of a transcriptional program in response to meiotic DNA breaks.**
**(A)** Western blot analysis of select transcriptionally induced genes. We used a 13xmyc tag to tag Rnr4 and Frd1, as antibodies for these genes were unavailable. Ponceau S staining of the membranes was the loading control. **(A, B)** Quantification of signals from Western blot analysis in (A) and additional replicate experiments. The plot shows the change of protein abundance compared with the 0-h time point for each sample group (*SPO11* or *spo11-YF*). **(C)** Northern blot analysis of select transcriptionally induced genes. *RNR4*, *FRD1*, and *PGK1* were induced at the RNA level >2-fold in *SPO11* cells compared with *spo11-YF* cells. *RAD51* was induced ~1.6-fold. Loading was normalized according to RNA concentrations measured after RNA extraction. **(C, D)** Quantification of the Northern blots in (C) and additional replicate experiments. The plot shows the change of mRNA abundance compared with the 0-h time point for each sample group (*SPO11* or *spo11-YF*). For n = 2 panels, we used the average of two biological replicates. Data in this figure and Fig S11 were from four independent time courses involving three different sets of *SPO11* and *spo11-YF* strains.

cells progressed into meiotic prophase (Figs 6C and D and S11A and B). mRNA abundance of vegetatively expressed transcripts, like *RNR4*, *FRD1*, and *PGK1*, remained low throughout the entire time course, and even transcripts that were initially induced upon meiotic entry, such as *RAD51* and *HOP1*, dropped in abundance once cells progressed further into meiotic prophase (Figs 6C and D and S11A and B). This decrease was observed regardless of sample normalization (Figs 6C and D and S11A and B) and was not a consequence of prophase arrest (Fig S11C) (Cheng et al, 2018). Nevertheless, transcript levels of the assayed genes were higher in *SPO11* cells compared with *spo11-YF* cells, confirming the induction of a transcriptional program in response to meiotic DNA breaks. However, this induction generally did not overcome the overall decrease in transcript abundance (Figs 6C and D and S11A and B). These data suggest that the decrease in transcript abundance following prophase entry dampens the effect of break-induced transcriptional changes sufficiently to cause only minor changes in protein levels.

## Discussion

In this study, we combined three systems-level approaches, that is, transcriptomics, proteomics, and phosphoproteomics, to capture

the breadth of the meiotic DNA break response in *S. cerevisiae*. Our analyses identified 332 DNA break–dependent phosphorylation events, substantially expanding our current knowledge of the meiotic DNA break response. The breadth of detection also high-lights the power of using complementary data acquisition tech-niques (DDA and DIA) and different analyses of the mass spectra (fold enrichment, presence/absence) for obtaining a large and high-quality dataset. Notably, the two approaches, fold enrichment and presence/absence, yielded qualitatively distinct groups of phosphorylation events. Fold enrichment analysis recovered many phosphorylation events that are not specific to meiosis, such as Hta1/2 S129 ($\gamma$-H2A) and Cbf1 S45, but are induced by different forms of canonical DNA damage signaling (Cobb et al, 2005; Smolka et al, 2007). In comparison, presence/absence analysis recovered most of the known meiosis-specific phosphorylation events, including Zip1 S75, Hed1 T40, and Rad54 T132, which therefore appear to be regulated in an on/off-switch like manner in response to meiotic DNA breakage. Thus, the meiotic DNA break response elaborates on features of the canonical DNA damage response by adding a large number of targets that specifically respond to meiotic DNA breaks.

Our dataset complements and expands on a published phos-phoproteomics analysis that compared strains with active or inactive Mek1 kinase (Suhandynata et al, 2016) by identifying targets of all DNA

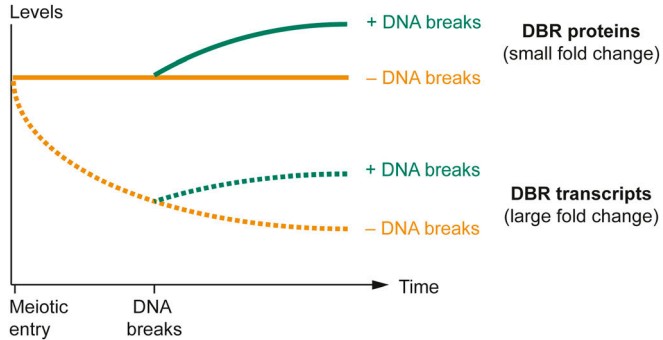

**Figure 7.  Model for dampened proteomic response to meiotic DNA breaks.** mRNA levels of many DNA break response (DBR) factors are highest at the time of meiotic entry supporting DBR protein production. As cells progress into meiotic prophase, mRNA levels of DBR factors (and other factors) drop but proteins remain at levels similar to the time of meiotic entry. DNA break formation (which initiates around 2 h after meiotic entry in our strain background) then triggers a transcriptional up-regulation of DBR factors. However, because of the earlier reduction in mRNA levels, even >2-fold up-regulation has only minor effects on proteins levels.

break–dependent kinases, and thus provides a comprehensive view of the phosphorylation events in response to DNA breaks. Importantly, our experimental setup blocked cells from exiting prophase and thus excluded cell-cycle-dependent changes in protein phosphorylation that occur once cells enter the meiotic divisions (Suhandynata et al, 2016). Our analyses therefore offer a snapshot of the immediate DNA break response before exit from meiotic prophase.

Surprisingly, we did not observe major proteome changes during meiotic DNA break formation. The observed concentrations of the >2,600 proteins (out of ~5,650 protein-coding genes in the yeast genome) correlated strongly between break-competent and break-defective cultures across three biological replicates (r = 0.99), and log fold-changes were narrowly distributed around 0. Although this analysis likely missed some low abundance proteins that may well undergo biologically important changes in abundance, the observed protein concentrations covered >4 orders of magnitude and therefore comprise a representative snapshot of the proteome. Notably, protein levels were steady even for genes that showed clear DNA break–dependent increases in mRNA abundance. Based on further inspection of candidate genes by immunoblotting and Northern analyses, we suggest that this dichotomy between steady protein levels and substantial transcriptional changes may be explained by low protein turnover after meiotic entry, combined with a steep drop in mRNA abundance that effectively dampens the effects of any subsequent break-dependent increases in mRNA levels (Fig 7). As a consequence, a twofold induction of mRNA levels has only negligible effects on protein levels. The steady protein abundances during meiotic prophase are consistent with the finding that most proteins are long-lived and that concentrations drop primarily as a consequence of dilution due to cell division (Martin-Perez & Villén, 2017), which does not occur during meiotic prophase. Targeted proteolysis via the meiosis-specific APC/Ama1 ubiquitin ligase or other mechanisms are important for meiotic prophase progression in yeast and mice (Kwon et al, 2003; Okaz et al, 2012), but our data do not suggest widespread differences in protein turnover in response to break formation. We observed large DNA break–dependent protein

abundance changes in only a few select proteins with specialized modes of regulation, including Sml1, whose proteolysis is triggered by phosphorylation and subsequent ubiquitination (Zhao & Rothstein, 2002) and Rad51, which is protected from degradation by phosphorylation (Woo et al, 2020).

The drop in transcript levels as cells enter the meiotic program may be related to the unique aspects of mRNA metabolism in meiotic prophase. For example, N6-Adenosine methylation of transcripts regulates both meiotic entry and meiotic commitment (Shah & Clancy, 1992; Agarwala et al, 2012; Bushkin et al, 2019). In addition, RNA stability is uncoupled from polyA-tail length during meiosis, likely through differential regulation of the RNA degradation protein Xrn1 (Wiener et al, 2021). Intriguingly, we observed DNA break dependent phosphorylation of Xrn1 on S1510, although the effect of this phosphorylation on Xrn1 activity remains to be determined.

Taken together, our results suggest extensive rewiring of the canonical DNA damage response in the context of meiotic DNA breaks to primarily use posttranslational signaling instead of major proteome changes. We speculate that this shift reflects the unique needs created by the programmed induction of nearly 200 DNA breaks. As protein modifications have the crucial potential to create spatially distinct and constrained signals that require comparatively little energy, they are uniquely suited to support the patterning of the meiotic recombination landscape and the creation of local or chromosome-wide dependencies in a shared nuclear environment, which is a key feature of meiotic recombination (Kar & Hochwagen, 2021).

# Materials and Methods

### Yeast strains and meiotic time courses

All strains used in this study were derived from the SK1 background. Table S3 lists the genotypes of these strains. Meiotic time courses were set up by growing cells at room temperature (25°C) in rich medium (YPD) for ~24 h, followed by inoculation at a final $OD_{600}$ of 0.3 in premeiotic BYTA medium (1% yeast extract, 2% bactotryptone, 1% potassium acetate, and 50 mM potassium phtalate) and growth for 16–17 h at 30°C. Cells were washed twice in sterile water and diluted to 1.9 $OD_{600}$ in SPO medium (0.3% potassium acetate) and sporulated at 30°C. The time of resuspension in SPO was defined as the 0 h time point.

### Flow cytometry analysis

We collected 150 μl of meiotic culture at the indicated time points and fixed cells with 350 μl 100% ethanol. The samples were stored at 4°C until further analysis. To prepare cells for flow cytometry, cell pellets were resuspended in 500 μl 50 mM Na-Citrate and treated with 0.7 μl RNAse A (20–40 mg/ml stock; Sigma-Aldrich) at 50°C for at least 1 d. 5 μl Proteinase K (20 mg/ml; VWR) was added to the samples and incubated at 50°C for at least 1 d before addition of 500 μl of 50 mM Na-Citrate with 0.1 μl SYTOX green (5 mM solution in DMSO; Invitrogen). Before cytometry, samples were sonicated for

~5 s at 10% amplitude. Signal was collected using a BD Accuri C6 Flow cytometer.

## Preparation of protein samples for mass spectrometry

We collected 50 ml meiotic culture at 5 h into meiosis, harvested cells at 4,000 rpm (Eppendorf Centrifuge 5810 R) for 3 min, and washed once with ice-cold sterile water at 4°C. Cell pellets were stored at −20°C until further processing. We used MS grade water in all the following steps and solutions. Proteins were extracted using an 8M urea lysis buffer (8M urea, 50 mM Tris–HCl, 75 mM NaCl, 1× Calbiochem protease inhibitor, 1 mM PMSF, 1× Thermo Fisher Scientific halt phosphatase, and protease inhibitor). 2× cell pellet volume of lysis buffer and 1× cell pellet volume of acid washed glass beads (Millipore Sigma) were added to 1.5 ml tubes, and samples were agitated with Digital Vortex Mixer (Thermo Fisher Scientific) at 4°C three times for 10 min separated by 2-min cooling intervals. The samples were centrifuged at 13,000 rpm (Eppendorf Centrifuge 5424) for 20 min and the supernatants were transferred to new 1.5-ml tubes. Protein concentrations were measured using a Quick Start Bradford Protein Assay (Bio-Rad) and 200 $\mu$g of protein was processed for preparation of WCE protein samples. Proteins were reduced in 5 mM DTT at 37°C for 30 min and alkylated in 15 mM iodoacetamide at room temperature (25°C) for 30 min in the dark. The alkylation was stopped by increasing the DTT concentration to 10 mM and incubating at room temperature for 15 min. Urea concentration was lowered by increasing the sample volume to 200 $\mu$l (sevenfold dilution) with 50 mM Tris–HCl pH 8 solution. We added 3 $\mu$g Trypsin Gold (Promega) to each sample for digestion and incubated samples at 37°C overnight (~16 h) while shaking. Digestion was stopped by adding formic acid to a final concentration of 1%. Samples were dried under vacuum until all liquid was removed and resuspended in Buffer C (95% water, 5% acetonitrile, 0.1% formic acid). HyperSep tips (Thermo Fisher Scientific) were used for clean-up following the kits instructions. Elution peptides were dried under vacuum until completely dry, resuspended in 100 $\mu$l amount of Buffer C, and stored at −80°C. Peptide concentrations were determined using Pierce Quantitative Fluorometric Peptide Assay (Thermo Fisher Scientific).

## Preparation of phospho-peptide enriched samples

We reduced 2,000 $\mu$g of protein samples and alkylated as described above. For phospho-peptide samples, the volume was increased to 2,000 $\mu$l (sevenfold dilution) and 30 $\mu$g Trypsin Gold (Promega) was added to each sample. After the samples were cleaned-up, phosphorylated peptides were enriched using the High-Select TiO$_2$ Phosphopeptide Enrichment Kit (Thermo Fisher Scientific) according to the manufacturer's instructions.

## Mass spectrometry analysis for DDA data

Samples were analyzed using an EASY-nLC 1,000 (Thermo Fisher Scientific) coupled to a QEHF instrument (Thermo Fisher Scientific). Peptides were separated using a PepMap C18 column (Thermo Fisher Scientific) with 155-min gradient of Buffer A (0.1% Formic Acid) and Buffer B (80% Acetonitrile, 0.1% Formic Acid). Full MS spectra were collected in a scan range of 375–1,500 with resolution of 120,000. AGC target was set to 3 × 10$^6$ and top 20 peptides were selected for further analysis with an isolation window of 1.5 m/z with a maximum injection time of 100 m/s. MS2 spectra were collected with a resolution of 30,000 and AGC target 2 × 10$^5$ with an isolation window of 1.5 m/z and normalized collision energy of 27 in centroid mode.

## Mass spectrometry analysis for DIA data

For DIA runs, a full MS scan was collected with a resolution of 120,000 and AGC target of 3 × 10$^6$ between 350 and 1,650 m/z. Each full MS scan was followed by 24 DIA windows with a resolution of 60,000 and AGC target of 1 × 10$^6$. Maximum injection time was set to auto, and normalized collision energy was set to 27 in profile mode. The DIA windows are provided in Table S4.

## Data analysis of DDA data

Raw data were processed in MaxQuant (version 1.5.5.1) (Tyanova et al, 2016) using the proteome of *S. cerevisiae* strain S288C (downloaded from Uniprot on 8 August 2017) with default settings. For phospho-enriched samples, Phospho(STY) was selected as a variable modification. Data were further analyzed and graphed in custom made R scripts using packages tidyverse, ggplot2 and limma (Ritchie et al, 2015; Wickham, 2016; Wickham et al, 2019). Proteins from contamination and reverse search were filtered out. For protein-level data, LFQ intensities were first normalized to parts per million and log$_2$ transformed. Log$_2$ transformed values were tested for significance analysis using limma. For analysis of phospho-proteome changes, we used the Phospho(STY) file. Phosphorylation-level data were filtered out to only include phosphosites with 0.9 localization probability. Then the phosphorylation site intensities were normalized to parts per million and log$_2$ transformed before significance analysis with limma.

## Data analysis of DIA data

DIA data were analyzed using Spectronaut (v 13.8.190930.43655) against a project-specific spectral library (phospho-DDA data). Perseus plug-in Peptide Collapse program was used to convert the Spectronaut (Biognosys) report file to intensity values at the peptide level (Bekker-Jensen et al, 2020). Peptide level intensity values were normalized to parts per million and log$_2$ transformed before performing significance analysis using limma.

## GSEA and motif analysis

Gene set enrichment analysis was performed using R package clusterProfiler using the enrichGO function (Yu et al, 2012). Motif analysis was performed with sequences of three amino acids on N-and C-terminal sides around the phosphosites using rmotif-x with *P*-value cut-off of 0.05 (Wagih et al, 2016). Sequences of all phosphosites detected in our study were used as the background dataset.

## Preparation of mRNA-Seq samples

### RNA extraction

We harvested 1.6 ml of meiotic culture at 4 h into meiosis and centrifuged samples at 3,000 rpm (Eppendorf Centrifuge 5424) for 5 min at 4°C. All supernatant was removed, and the pellet was resuspended with 1 ml of Tris–EDTA (10 mM Tris, pH 8, 1 mM EDTA) buffer. The samples were spun down again, and the supernatant was removed. The samples were stored at −80°C until further processing. RNA extraction was performed using the RNeasy Mini Kit (QIAGEN). 600 $\mu$l of RLT buffer with 1% (vol/vol) $\beta$-mercaptoethanol and ~200 mg glass beads were added to the pellets. The samples were agitated for 20 min at 4°C and spun down at 14,000 rpm (Eppendorf Centrifuge 5424) for 2 min. Supernatant was transferred to a new microcentrifuge tube and mixed 1:1 with 70% ethanol. Samples were transferred to RNeasy columns and RNA extraction was completed following the kit's instructions. We measured the RNA integrity using Agilent RNA ScreenTape and RNA concentration using Qubit RNA HS assay kit (Thermo Fisher Scientific). We used 1.2 $\mu$g total RNA for mRNA purification. mRNAs were purified using Sera-Mag oligo(dt) magnetic particles (Sigma-Aldrich). mRNAs were fragmented with Ambion mRNA fragmentation buffer and fragmented mRNAs were purified using RNeasy MinElute Kit (QIAGEN). Final elution volume was 9 $\mu$l.

### First- and second-strand synthesis

First-strand synthesis was performed in a similar manner as described in Parkhomchuk et al (2009). For first-strand synthesis, 8 $\mu$l fragmented mRNAs were mixed with 1 $\mu$l of random hexamers (Invitrogen) and 1 $\mu$l of 10 mM dNTPs. The samples were incubated at 65°C for 5 min and chilled on ice for 1 min. We added 10 $\mu$l of a master mix with final concentrations of 1× RT Buffer (Thermo Fisher Scientific), 10 mM MgCl$_2$, 20 mM DTT, 4 U/$\mu$l RnaseOUT (Thermo Fisher Scientific), and 20 U/$\mu$l of SuperScript III RT (Thermo Fisher Scientific) to the RNA samples. The samples were then first incubated at 25°C for 10 min, followed by a 50-min incubation at 50°C. The reaction was stopped by incubating samples at 75°C for 15 min. For dNTP cleanup, 80 $\mu$l water, 1 $\mu$l glycogen, 10 $\mu$l 3 M NaOAc (pH 5.2), and 200 $\mu$l cold ethanol were added to the samples. Samples were stored at −80°C for 3–7 d. Samples were centrifuged at 14,000 rpm (Eppendorf Centrifuge 5424) for 20 min at 4°C. Supernatant was removed and 500 $\mu$l of cold 75% ethanol was added to the samples. Samples were centrifuged again at 14,000 rpm (Eppendorf Centrifuge 5424) for 10 min at 4°C. Supernatant was removed and samples were resuspended in a mixture composed of 51 $\mu$l RNAse-free water, 1 $\mu$l of 10× RT buffer, 1 $\mu$l 100 mM DTT, 2 $\mu$l of 25 mM MgCl$_2$. Second-strand synthesis was performed as described in Parkhomchuk et al (2009).

### Library preparation

Library preparation was performed using TruSeq Library prep kit v1, but the adapters were used at 1:20 and 1U of UNG enzyme (Thermo Fisher Scientific) was added before PCR amplification to digest uridine containing templates to produce directional libraries. For uridine digestion, the samples were incubated at 37°C for 15 min and the reaction was terminated by incubation at 98°C for 10 min. Amplified DNA was run on a 1.5% agarose gel and DNA between 250

bp and 600 bp was extracted using a QIAGEN Gel Extraction kit with a MiniElute column. DNA concentrations were measured using Qubit dsDNA HS Assay Kit (Thermo Fisher Scientific) and KAPA library quantification kit (Roche). DNA sizes were checked with Agilent High Sensitivity D1000 ScreenTape. 75-bp pair-ended sequencing was performed on a NextSeq 500 instrument.

## Analysis of RNA-Seq samples

RNA-Seq reads were mapped to the SK1 genome using the nf-core RNA-Seq pipeline (Yue et al, 2017; Ewels et al, 2020; Patel et al, 2021). We used the salmon.merged.gene_counts.rds file from salmon output for further analysis. Combat-Seq was used for batch correction and DeSeq2 was used for principal components analysis and for differential gene expression analysis (Love et al, 2014; Zhang et al, 2020).

## Immunoblotting

For immunoblotting, 5 ml samples were collected at the indicated time points. The samples were spun down at 2,500 rpm (Eppendorf Centrifuge 5810 R) for 2.5 min, and pellets were resuspended in 5% TCA. The samples were kept on ice for at least 10 min after resuspension. The samples were washed with 500 $\mu$l 1 M Tris and resuspended in 80 $\mu$l of TE+DTT (0.8XTE, 200 mM DTT) buffer. After the addition of 30 $\mu$l 5× SDS buffer (190 mM Tris-acetate, 6% $\beta$-mercaptoethanol, 30% glycerol, 20% SDS, and 0.05% bromophenol blue), the samples were incubated at 100°C for 5 min and stored at −80°C immediately. Proteins were run in hand-cast 10% 29:1 (acrylamide: bis-acrylamide) gels for Hrr25 blots and hand-cast 8% 29:1 (acrylamide: bis-acrylamide) gels for Rad53 blots. 4–20% gradient gels (Bio-Rad) were used for Sml1 blots and 4–15% gradient gels (Bio-Rad) were used for Frd1-13myc, Rnr4-13myc, and Rad51 blots. For immunoblotting experiments with linear range lanes, proteins were run using an Owl vertical electrophoresis system (model P10DS) with hand-cast 10% acrylamide:Bis (37.5:1) gels. Transfer was performed with Owl semi-dry transfer system. All blots were blocked with 5% milk. Primary antibodies were used at the following concentrations: Hrr25 ph-S438 (rabbit, 1:1,000), Rad53 (goat, yc-19 Santa Cruz) at 1:500, Sml1(rabbit, AgriSera) at 1:1,000, and $\beta$-actin (rabbit, CST) at 1:1,000, Myc-tag at 1:1,000 (rabbit, CST). Anti-rabbit secondary antibody (Kindle Biosciences) was used at 1:2,000, anti-goat secondary antibody (Kindle Biosciences) was used at 1:1,000. The blots were visualized using KwikQuant Imager (Kindle Biosciences). The phospho-specific Hrr25 ph-S438 antibody was raised by Covance against the synthetic target peptide Ac-QQRD(pS)QEQQC-amide.

## Northern blotting

For Northern blotting, 6 ml samples were collected at the indicated time points, spun down at 2,500 rpm (Eppendorf Centrifuge 5810 R) for 2.5 min and stored in 2 ml microcentrifuge tubes at −80°C immediately. Pellets were overlaid with 350 $\mu$l acid phenol chloroform pH4.5 (Thermo Fisher Scientific). After addition of 100 mg glass beads and 350 $\mu$l RNA buffer 1 (300 mM NaCl, 10 mM Tris–HCl, pH 6.8, 1 mM EDTA, and 0.2% SDS) samples were agitated at 4°C for 10 min in a Disruptor Genie (Scientific Instruments). Phases were

separated by centrifuging 10 min at 14,000 rpm (Eppendorf Centrifuge 5424) at 4°C and 300 $\mu$l of the aqueous phase were precipitated in 1 ml cold 100% ethanol at 4°C for 10 min. RNA was collected by centrifuging for 5 min at 14,000 rpm (Eppendorf Centrifuge 5424) at 4°C and pellets were resuspended in RNA buffer 2 (10 mM Tris–HCl, pH 6.8, 1 mM EDTA, and 0.2% SDS) at 65°C for 20 min before storing at at –20°C. RNA concentration was determined using a NanoDrop instrument. Samples were denatured for 10 min at 65°C in denaturation mix (40 mM MOPS, pH 7.0, 50% formamide, and 6.5% formaldehyde) and separated in a 1.1% agarose gel containing 6.2% formaldehyde and 40 mM MOPS, pH 7.0. RNA was blotted onto a HybondN+ membrane using neutral transfer in 10× SSC and UV cross-linked. Radioactive probes were synthesized from gel-purified templates using a Prime-it RmT Random Labeling Kit (Agilent) and $\alpha$-$^{32}$P-dCTP (Perkin Elmer). Templates were produced by PCR using the following primers (*RNR4*: F 5′-CAG CCG TAG ATT CGT GAT GTT CCC-3′, R 5′-GCG GAC TTA GAC ATG TCA CTG GCC-3′; *FRD1*: F 5′-GGT TTG GCC GGG CTG GCT GC-3′, R 5′-GCA TAA TTG GGC GAC AGT GAT TGG-3′; *RAD51*: F 5′-CAG CTT CAG TAC GGG AAC GGT TCG-3′, R 5′-GCC ATA CCA CCA TCA ACT TGG GCG-3′; *HOP1*: F 5′-CCC AAT CCC TGG AAC CTT TAC CCC-3′, R 5′-GCT CCT GTA GGG TTG ACG ACG GAG-3′; *PGK1*: F 5′-TGA CTT CAA CGT CCC ATT GGA CGG-3′, R 5′-AAC ACC TGG ACC GT CCA GAC-3′). Signals were measured using a Typhoon FLA9000 instrument.

## Data Availability

All mass spectrometry data are deposited at the PRIDE database PXD031779 and PXD031781 (Perez-Riverol et al, 2022). RNA-seq data are deposited at the GEO database GSE197022.

## Supplementary Information

## Acknowledgements

A Hochwagen acknowledges funding by the US National Institutes of Health (R01GM111715 and R01GM123035). C Vogel acknowledges funding by the US National Institutes of Health (R35GM127089/NH/NIH HHS/United States). FM Kar acknowledges support from the Chair's Graduate Fellowship (NYU). We thank the NYU Department of Biology Sequencing Core for technical assistance and data processing.

### Author Contributions

FM Kar: conceptualization, software, formal analysis, validation, investigation, visualization, methodology, and writing—original draft, review, and editing.
C Vogel: conceptualization, supervision, funding acquisition, investigation, and writing—original draft, review, and editing.
A Hochwagen: conceptualization, formal analysis, supervision, funding acquisition, investigation, and writing—original draft, review, and editing.

### Conflict of Interest Statement

The authors declare that they have no conflict of interest.

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
