## [Reviewer comments · Life Science Alliance]

Life Science Alliance

Meiotic DNA breaks activate a streamlined kinase response that largely avoids protein level changes

Funda Kar, Christine Vogel, and Andreas Hochwagen

DOI: <https://doi.org/10.26508/lsa.202201454>

Corresponding author(s): *Andreas Hochwagen, New York University and Christine Vogel, New York University*

Review Timeline:

Submission Date:	2022-03-17
Editorial Decision:	2022-04-14
Revision Received:	2022-08-10
Editorial Decision:	2022-08-12
Revision Received:	2022-08-15
Accepted:	2022-08-19

Transaction Report:

April 14, 2022

Re: Life Science Alliance manuscript #LSA-2022-01454-T

Prof Andreas Hochwagen
New York University
Department of Biology
1009 Silver Center 100 Washington Square East
New York, NY 10003

Dear Dr. Hochwagen,

Thank you for submitting your manuscript entitled "Meiotic DNA breaks activate a streamlined kinase response that largely avoids protein level changes" to Life Science Alliance. The manuscript was assessed by expert reviewers, whose comments are appended to this letter. We invite you to submit a revised manuscript addressing the Reviewer comments.

Thank you for this interesting contribution to Life Science Alliance. We are looking forward to receiving your revised manuscript.

Sincerely,

Eric Sawey, PhD
Executive Editor
Life Science Alliance
<http://www.lsa-journal.org>

B. MANUSCRIPT ORGANIZATION AND FORMATTING:

Reviewer #1 (Comments to the Authors (Required)):

This manuscript pursues an interesting question: What are the effects of the meiotic DNA breaks that initiate recombination on phosphorylation and gene expression? Kinases, including ATR, ATM, and CHK2 have known important roles in meiosis and in the mitotic DNA damage response, and therefore identifying the targets that they phosphorylate in meiosis is important and may enable future mechanistic studies of the meiotic DNA break checkpoint. The authors identify hundreds of new sites using a phospho proteomic approach, and verify one new site on HRR25 that could be exciting for further study. This section of the paper is important and well founded. These data will be an asset to the community.

The authors then turn to transcriptomics and proteomics to identify 23 genes for which a 2-fold transcriptional difference is seen between wild type and DSB deficient cells but no proteomic change is detected. This is the basis for the strong statement that, excepting four proteins, protein abundances do not change in response to DSB formation despite transcript abundance changes. This section of the manuscript is problematic. More below.

1. The proteomic and transcriptomic data is not available. The authors state that it is deposited and provide accession information but I could not access any of the three datasets. This is tremendously problematic. It limits the ability to review the manuscript rigorously.
2. From the data provided, it seems that the authors are reporting a detection difference between mRNA and protein abundance changes. This is not equivalent to no biological change in mRNA and protein abundances, which cannot be claimed from these data. Comparing fold changes between proteomic data and transcriptomic data is difficult, in part because of major differences in the linearity of detection for the two approaches. Unlike sequencing data, observing no change in protein levels by mass spectrometry only means that no change is detected by these data analyzed in this way, not that no change exists. At points, the authors imply this but they also state that "the meiotic DNA break response does not involve substantial changes in translation or protein degradation" (page 8 line 294-295) and that the proteome is "largely stable" (abstract), as two of several examples. The former statement has no support from the data shown and the second is misleading. Translation and protein degradation are not measured. It is possible that degradation and translation are occurring but that the net result is low protein abundance changes. It is also possible that moderate protein level changes that cannot be detected using their approach are biologically important. What are the fold change limits of the protein abundance detection used? What fold changes are substantial and how is this determined?
3. It is possible that the proteins that show the biggest changes are among the half of the proteome that they do not detect and thus the set of proteins measured may not be representative of a global trend. This must be discussed.
4. Fundamental differences in the stability of mRNA compared to protein make fold changes in a set time period incomparable. The authors have collected time resolved data and the data trends through time. Presenting these measurements would be more convincing than the fold change analyses shown here, especially because it is not obvious what the fold change refers to in all cases (range of change over the entire time period or difference between 0 and 5 hours?). Figure 5E shows mRNA fold changes. The same plots for protein for positive controls and the cases that are stated to have no change would be useful.
5. Is the authors' model that low mRNA levels mean that fold changes still cause low mRNA, and so the translation from this small amount of mRNA does not change steady state protein levels by a large fold change? Or is there truly no change in protein abundances? The latter model would require a way to silence mRNA translation, or balanced synthesis and degradation. The former model is not surprising based on the properties of mRNA and protein stability. An exception would be if the mitotic DSB response leads to similar fold changes in protein and mRNA abundance, but this is not stated. Whatever the model, it should be discussed in a clear way throughout the manuscript and should accurately represent the data and its limitations.

Minor additional points:

1. Western blotting to confirm no protein level changes requires replicates and quantification, and controls to confirm the linear detection range, in Figure 5C.
2. It is stated that there was high reproducibility for the phospho data (Line 120) but the Figure S2 data do not support this. R values are low and replicate patterns are not similar between replicates. High reproducibility may not be needed to identify phosphorylation sites but this should be stated accurately.

Reviewer #2 (Comments to the Authors (Required)):

The manuscript entitled "Meiotic DNA breaks activate a streamlined kinase response that largely avoids protein level changes" by Kar FM et al. provides a very robust, detailed and well described data set of proteome changes upon meiotic DNA breaks in yeast. This study is of high importance as the understanding of phosphosignaling pathways during meiotic events is just emerging. The proteomic study is well controlled. Biological replicates and technical variations have been carefully analyzed, and the methods are described in detail. The authors observed that upon meiotic events the proteome was surprisingly stable, when evaluating protein levels. Considering the phosphoproteome, previously known and novel phosphorylation events were described, which could be partially attributed to specific kinases. Other kinases will be identified by future studies, which will build upon the data of the current study. In summary, I trust that this study is a very valuable contribution for the journal. Therefore, I recommend publication of this work.

My cross-comments to Reviewer 1:

"The proteomic and transcriptomic data is not available."

I agree with Reviewer 1 that these data must be provided.

We would like to thank the editor and reviewers for their overall enthusiasm and for their thoughtful suggestions that have helped to further improve this manuscript. We have fully addressed all points that were raised.

Reviewer #1 (Comments to the Authors (Required)):

This manuscript pursues an interesting question: What are the effects of the meiotic DNA breaks that initiate recombination on phosphorylation and gene expression? Kinases, including ATR, ATM, and CHK2 have known important roles in meiosis and in the mitotic DNA damage response, and therefore identifying the targets that they phosphorylate in meiosis is important and may enable future mechanistic studies of the meiotic DNA break checkpoint. The authors identify hundreds of new sites using a phospho proteomic approach, and verify one new site on HRR25 that could be exciting for further study. This section of the paper is important and well founded. These data will be an asset to the community.

The authors then turn to transcriptomics and proteomics to identify 23 genes for which a 2-fold transcriptional difference is seen between wild type and DSB deficient cells but no proteomic change is detected. This is the basis for the strong statement that, excepting four proteins, proteins abundances do not change in response to DSB formation despite transcript abundance changes. This section of the manuscript is problematic. More below.

1. The proteomic and transcriptomic data is not available. The authors state that is deposited and provide accession information but I could not access any of the three datasets. This is tremendously problematic. It limits the ability to review the manuscript rigorously.

We sincerely apologize for this oversight. Reviewer access information is:

--GEO accession GSE197022

Reviewer token: onursciqblkhpqz

--PRIDE accession PXD031779

Username: reviewer_pxd031779@ebi.ac.uk

Password: LKXoLmc8

--PRIDE accession PXD031781

Username: reviewer_pxd031781@ebi.ac.uk

Password: rqnTiwUh

2. From the data provided, it seems that the authors are reporting a detection difference between mRNA and protein abundance changes. This is not equivalent to no biological change in mRNA and protein abundances, which cannot be claimed from these data. Comparing fold changes between proteomic data and transcriptomic data is difficult, in part because of major

differences in the linearity of detection for the two approaches. Unlike sequencing data, observing no change in protein levels by mass spectrometry only means that no change is detected by these data analyzed in this way, not that no change exists. At points, the authors imply this but they also state that "the meiotic DNA break response does not involve substantial changes in translation or protein degradation" (page 8 line 294-295) and that the proteome is "largely stable" (abstract), as two of several examples. The former statement has no support from the data shown and the second is misleading. Translation and protein degradation are not measured. It is possible that degradation and translation are occurring but that the net result is low protein abundance changes. It is also possible that moderate protein level changes that cannot be detected using their approach are biologically important. What are the fold change limits of the protein abundance detection used? What fold changes are substantial and how is this determined?

We thank the reviewer for highlighting these points. We carefully rewrote the relevant passages to clarify the types of measurements conducted in the study and the interpretation of the lack of changes observed.

We further clarified in the manuscript that our results suggest that there is no "large-scale remodeling" of the proteome. This conclusion was based on the exceedingly high correlation between the proteomes of break-defective and break-competent cultures, which showed an r value of 0.99 in three biological replicates.

We note that quantitative mass spectrometry-based proteomics is a well-established method that can detect very small fold-changes (<2-fold) with high statistical confidence. The revised manuscript lists evidence for the accuracy of the proteomics measurements:

- i) Leu2 serves as a positive control, as it is expected to be 2-fold more abundant in *SPO11* cells (four copies) than in the *spo11-YF* cells (two copies). Indeed, the mass spectrometry data reports Leu2 as one of the two maximally changed proteins between the cell types, supporting the sensitivity of the measurements.
- ii) If quantitative measurements were random, we would observe differential expressed proteins between replicates or between individual samples. This differential expression would occur just by chance and would vary across replicates. This is not the case: measurements between replicates are highly consistent, as Supplementary Figure 7 shows.
- iii) We verified the mass spectrometry-based results for select proteins using an orthogonal method. We specifically tested proteins known to change during the mitotic DNA damage response. Western blotting confirmed that the proteins essentially do not change during the meiotic DNA damage response.
- iv) We examined the proteins with changes at the RNA level but not observed protein abundance changes more closely and added this analysis to Supplementary Figure 10. We found that the proteins' abundances changed very minimally: genes that showed >2-fold changes by mRNA-seq, showed <1.1-fold change in protein levels, a change that is statistically significant (when compared to the whole proteome by bootstrapping analysis). These small changes are consistent with our model that the

initial drop in mRNA levels effectively dampens the effect that relatively high fold changes in mRNA levels have on the proteome, as we now illustrate in a model figure (Figure 7).

3. It is possible that the proteins that show the biggest changes are among the half of the proteome that they do not detect and thus the set of proteins measured may not be representative of a global trend. This must be discussed.

We revised the manuscript to clarify that we recovered a very large fraction of the expected proteome. Although the yeast genome contains about 5,650 protein-coding genes, there is no expectation that all are expressed during meiosis. Like higher eukaryotes, yeast cells undergo very specific developmental transitions that are defined by the induction and repression of large sets of genes. Thus, by quantifying >2,600 proteins, with concentrations ranging >4 orders of magnitude, we likely captured much more than half of the expressed proteome of meiotic prophase. Although we cannot exclude that our analysis missed a number of protein changes, we deem it highly unlikely that our analyses would have missed a global trend. We rewrote the relevant section of the discussion to distinguish between global trends, which we would be able to detect, and individual protein changes, which we may have missed.

4. Fundamental differences in the stability of mRNA compared to protein make fold changes in a set time period incomparable. The authors have collected time resolved data and the data trends through time. Presenting these measurements would be more convincing than the fold change analyses shown here, especially because it is not obvious what the fold change refers to in all cases (range of change over the entire time period or difference between 0 and 5 hours?). Figure 5E shows mRNA fold changes. The same plots for protein for positive controls and the cases that are stated to have no change would be useful.

We revised the manuscript to include time-resolved western blot data for Rad51, Frd1, Rnr4, and Pgc1 to compare the observed protein level changes to the observed mRNA level changes, shown in Figure 6. Again, the plots confirm the results gained from the other experiments.

5. Is the authors' model that low mRNA levels mean that fold changes still cause low mRNA, and so the translation from this small amount of mRNA does not change steady state protein levels by a large fold change? Or is there truly no change in protein abundances? The latter model would require a way to silence mRNA translation, or balanced synthesis and degradation. The former model is not surprising based on the properties of mRNA and protein stability. An exception would be if the mitotic DSB response leads to similar fold changes in protein and mRNA abundance, but this is not stated. Whatever the model, it should be discussed in a clear way throughout the manuscript and should accurately represent the data and its limitations.

We clarified our interpretation and model using Figure 7: initial mRNA drop dampens the effect of the transcriptional response, and a two-fold change in RNA abundance does not lead to a two-fold change in protein concentrations. Similar to what the reviewer described: as the cells

start with high protein concentrations, low mRNA levels, even though they increase several fold, do hardly affect protein concentrations. The analysis outlined in response to point #2 (iv) confirms this interpretation. We included additional discussion of these findings, the model, and the comparison to the mitotic DNA damage response in the revised manuscript.

Minor additional points:

1. Western blotting to confirm no protein level changes requires replicates and quantification, and controls to confirm the linear detection range, in Figure 5C.

We added the western blot data showing that signals are not saturated and in the linear range to Supplemental Figure 10. The results confirm the observations from the mass spectrometry data.

2. It is stated that there was high reproducibility for the phospho data (Line 120) but the Figure S2 data do not support this. R values are low and replicate patterns are not similar between replicates. High reproducibility may not be needed to identify phosphorylation sites but this should be stated accurately.

We added revised the manuscript to clarify that high reproducibility was found for data points in the upper-right quadrant, i.e. the set of true DNA break-dependent phosphorylation events that appear in the break-competent SPO11 strain but not in the spo11-YF mutant. The correlation between replicates is lower if all data point, including false positive events, are included.

Reviewer #2 (Comments to the Authors (Required)):

The manuscript entitled "Meiotic DNA breaks activate a streamlined kinase response that largely avoids protein level changes" by Kar FM et al. provides a very robust, detailed and well described data set of proteome changes upon meiotic DNA breaks in yeast. This study is of high importance as the understanding of phosphosignaling pathways during meiotic events is just emerging. The proteomic study is well controlled. Biological replicates and technical variations have been carefully analyzed, and the methods are described in detail. The authors observed that upon meiotic events the proteome was surprisingly stable, when evaluating protein levels. Considering the phosphoproteome, previously known and novel phosphorylation events were described, which could be partially attributed to specific kinases. Other kinases will be identified by future studies, which will build upon the data of the current study. In summary, I trust that this study is a very valuable contribution for the journal. Therefore, I recommend publication of this work.

We thank the reviewer for their kind comments.

August 12, 2022

RE: Life Science Alliance Manuscript #LSA-2022-01454-TR

Prof. Andreas Hochwagen
New York University
Department of Biology
24 Waverly Place
6th floor
New York, NY 10003

Dear Dr. Hochwagen,

Thank you for submitting your revised manuscript entitled "Meiotic DNA breaks activate a streamlined kinase response that largely avoids protein level changes". We would be happy to publish your paper in Life Science Alliance pending final revisions necessary to meet our formatting guidelines.

- please address Reviewer 1's final comment
- please add ORCID ID for secondary corresponding author-they should have received instructions on how to do so
- please add the Twitter handle of your host institute/organization as well as your own or/and one of the authors in our system
- please use the [10 author names, et al.] format in your references (i.e. limit the author names to the first 10)
- please make the deposited datasets publicly-accessible at this point, and you can remove the Reviewer access info from your Data Availability section (thank you for providing)
- please add sizes next to all blots

A. FINAL FILES:

B. MANUSCRIPT ORGANIZATION AND FORMATTING:

Sincerely,

Reviewer #1 (Comments to the Authors (Required)):

The authors have greatly improved the manuscript with added data and clarified writing. The clarified description of results is important and that it should be extended to the abstract for the following sentence: "We also observed a clear transcriptional program, but only very few changes in protein levels." If this read "We detected a clear transcriptional program, but detected only very few changes in protein levels" I would fully support publication of the revised manuscript.

August 19, 2022

RE: Life Science Alliance Manuscript #LSA-2022-01454-TRR

Prof. Andreas Hochwagen
New York University
Department of Biology
24 Waverly Place
6th floor
New York, NY 10003

Dear Dr. Hochwagen,

Thank you for submitting your Research Article entitled "Meiotic DNA breaks activate a streamlined kinase response that largely avoids protein level changes". It is a pleasure to let you know that your manuscript is now accepted for publication in Life Science Alliance. Congratulations on this interesting work.

DISTRIBUTION OF MATERIALS:

Again, congratulations on a very nice paper. I hope you found the review process to be constructive and are pleased with how the manuscript was handled editorially. We look forward to future exciting submissions from your lab.

Sincerely,
